# Antimicrobial Resistance in the Aconcagua River, Chile: Prevalence and Characterization of Resistant Bacteria in a Watershed Under High Anthropogenic Contamination Pressure

**DOI:** 10.3390/antibiotics14070669

**Published:** 2025-07-02

**Authors:** Nicolás González-Rojas, Diego Lira-Velásquez, Richard Covarrubia-López, Johan Plaza-Sepúlveda, José M. Munita, Mauricio J. Carter, Jorge Olivares-Pacheco

**Affiliations:** 1Grupo de Resistencia Antimicrobiana en Bacterias Patógenas y Ambientales, (GRABPA), Instituto de Biología, Pontificia Universidad Católica de Valparaíso, Valparaíso 2340025, Chile; nicolas.gonzalez.r@mail.pucv.cl (N.G.-R.); diego.lira@pucv.cl (D.L.-V.); richard.covarrubia@pucv.cl (R.C.-L.); johan.plaza.s@mail.pucv.cl (J.P.-S.); 2Multidisciplinary Initiative for Collaborative Research on Bacterial Resistance (MICROB-R), Santiago 7610658, Chile; josemunita@udd.cl; 3Genomics & Resistant Microbes Group (GeRM), Instituto de Ciencias e Innovación en Medicina (ICIM), Facultad de Medicina Clínica Alemana, Universidad del Desarrollo, Santiago 7610658, Chile; 4Onw Health Institute, Facultad de Ciencias de la Vida, Universidad Andrés Bello, Santiago 7591538, Chile; mauricio.carter@unab.cl; 5Departamento de Ecología, Center of Applied Ecology and Sustainability (CAPES), Facultad de Ciencias Biológicas, Universidad Católica de Chile, Santiago 8330009, Chile

**Keywords:** Aconcagua river, antibiotic-resistant bacteria (ARB), One Health, Environmental *Pseudomonas*, *bla_VIM_*

## Abstract

**Background**: Antimicrobial resistance (AMR) is a growing global health concern, driven in part by the environmental release of antimicrobial-resistant bacteria (ARB) and antimicrobial resistance genes (ARGs). Aquatic systems, particularly those exposed to urban, agricultural, and industrial activity, are recognized as hotspots for AMR evolution and transmission. In Chile, the Aconcagua River—subject to multiple anthropogenic pressures—offers a representative model for studying the environmental dimensions of AMR. **Methods:** Thirteen surface water samples were collected along the Aconcagua River basin in a single-day campaign to avoid temporal bias. Samples were filtered through 0.22 μm membranes and cultured on MacConkey agar, either unsupplemented or supplemented with ceftazidime (CAZ) or ciprofloxacin (CIP). Isolates were purified and identified using MALDI-TOF mass spectrometry. Antibiotic susceptibility was evaluated using the Kirby–Bauer disk diffusion method in accordance with CLSI guidelines. Carbapenemase activity was assessed using the Blue-Carba test, and PCR was employed for the detection of the *bla_VIM_*, *bla_KPC_*, *bla_NDM_*, and *bla_IMP_* genes. **Results:** A total of 104 bacterial morphotypes were isolated; 80 were identified at the species level, 5 were identified at the genus level, and 19 could not be taxonomically assigned using MALDI-TOF. *Pseudomonas* (40 isolates) and *Aeromonas* (25) were the predominant genera. No growth was observed on CIP plates, while 24 isolates were recovered from CAZ-supplemented media, 87.5% of which were resistant to aztreonam. Five isolates exhibited resistance to carbapenems; two tested positive for carbapenemase activity and carried the *bla_VIM_* gene. **Conclusions:** Our results confirm the presence of clinically significant resistance mechanisms, including *bla_VIM_*, in environmental *Pseudomonas* spp. from the Aconcagua River. These findings highlight the need for environmental AMR surveillance and reinforce the importance of adopting a One Health approach to antimicrobial stewardship and wastewater regulation.

## 1. Introduction

Antimicrobial resistance (AMR) represents one of the most critical threats to global public health, associated with high morbidity and mortality rates [1,2]. It is estimated to cause around 700,000 deaths annually, a number that could rise to 10 million by the year 2050 in the absence of effective interventions [3]. This situation not only compromises the efficacy of antibacterial treatments but also generates a considerable economic impact by prolonging hospital stays and increasing healthcare costs [4,5]. Although antimicrobial-resistant bacteria (ARB) and antimicrobial-resistance genes (ARGs) are naturally present in microbial ecosystems, most of them are not clinically relevant [6,7,8]. However, human activity has favored the introduction and dissemination of clinically important ARB and ARGs in environments where they were previously absent, turning them into emerging contaminants [9,10]. This dissemination is exacerbated by the massive release of antibiotic residues (ARs), which exert selective pressure that promotes the persistence and horizontal transfer of clinically relevant ARGs [11,12]. Thus, clinically relevant ARs, ARB, and ARGs converge as an environmental threat, as they facilitate the evolution of resistance and its spread among environmental and pathogenic microbial communities, significantly altering the microbial ecology of affected ecosystems [13,14,15]

On the other hand, aquatic environments have been identified as key reservoirs of AMR, as they receive contaminants from various sources such as urban wastewater, agricultural runoff, and industrial discharges [13,14]. In these systems, genetic interactions occur between environmental and human-associated pathogenic bacteria, contributing to the emergence and transfer of resistance mechanisms [16,17]. Several studies have demonstrated the persistence of ARGs and ARB in water bodies used for human consumption, irrigation, and recreation, increasing the risk of direct and indirect exposure to multidrug-resistant (MDR) pathogens [13]. In this context, the Aconcagua River—located in the Mediterranean region of central Chile—represents a model system for the study of AMR in aquatic environments. Its flow depends mainly on seasonal snowmelt from the Andes Mountains, and its basin is subject to multiple anthropogenic pressures. Intensive agricultural activity, including the use of fertilizers and pesticides, exerts selective pressure on bacterial communities, while mining activities contribute tailings rich in heavy metals, which can co-select for antibiotic resistance determinants [18,19,20]. This is further compounded by urban influences, as at least four cities with populations exceeding 50,000 inhabitants (including Quillota, San Felipe, Los Andes, and La Calera) discharge their treated wastewater into the river. These treatment plants comply solely with current Chilean regulations, which do not include microbiological criteria—that is, they do not establish a maximum allowable concentration of microorganisms in the treated water to validate the efficacy of the treatment—thus permitting the potential release of ARB and ARGs into the environment. Considering that more than 370,000 people influence the water quality of the Aconcagua River directly or indirectly and that this aquatic system concentrates multiple factors relevant to the evolution and dissemination of AMR, its study offers a unique opportunity. Moreover, its accessibility and the possibility of conducting comprehensive sampling in a single day allow for the minimization of temporal biases. In this study, we propose the isolation and characterization of ARB present in the Aconcagua River, with the aim of understanding their diversity and the impact of human activities on the spread of AMR in natural environments.

## 2. Results

### 2.1. Isolation and Identification of Bacterial Isolates

A total of 104 bacterial morphotypes were identified and isolated from the collected samples. Of these, 83 were recovered from control media (MacConkey without antibiotics), and 24 were recovered from media supplemented with ceftazidime (CAZ). No growth was observed on the plates supplemented with ciprofloxacin (CIP). Taxonomic identification using MALDI-TOF enabled the identification of 80 isolates at the species level (Score ≥ 2.000) and 5 just at the genus level (Score 1.999–1.700). Nineteen isolates could not be identified by MALDI-TOF. The predominant genus was Pseudomonas (40 isolates), followed by Aeromonas (25), Comamonas (3), Rahnella (3), Pantoea (3), and Kosakonia (2). The remaining genera were each represented by a single isolate (Figure 1).

### 2.2. Antimicrobial Susceptibility Profile

Analysis of the susceptibility profiles revealed that 87.5% (21/24) of the isolates from CAZ-supplemented media were resistant to aztreonam (ATM). Additionally, 25% (6/24) were non-susceptible to ceftazidime (CAZ), including four resistant and two intermediately susceptible isolates (Figure 2). For cefepime (FEP), 8.3% (2/24) of isolates were non-susceptible—one resistant and one intermediate. Resistance to imipenem (IMP) and meropenem (MEM) was observed in 20.8% (5/24) and 12.5% (3/24) of isolates, respectively. Notably, three isolates showed resistance to both carbapenems. Resistance to ceftriaxone (CRO) and chloramphenicol (C) was detected in 4.2% (1/24) of isolates. Intermediate susceptibility to trimethoprim-sulfamethoxazole (SXT) was also observed in 4.2% (1/24) of isolates. Gentamicin (CN) resistance was found in 8.3% (2/24) of isolates, both of which were also resistant to carbapenems. All isolates remained susceptible to ciprofloxacin (CIP), levofloxacin (LEV), amikacin (AK), and piperacillin/tazobactam (TZP). None of the isolates met the criteria for classification as multidrug-resistant (MDR), extensively drug-resistant (XDR), or pandrug-resistant (PDR) (Figure 3A,B). In terms of geographic distribution, sampling point 8 yielded the highest number of resistant isolates (7/24), followed by point 5 (5/24). Among the five carbapenem-resistant isolates, three—resistant to both IMP and MEM—were recovered from point 3, while the remaining two—resistant only to IMP—were isolated from points 4 and 8.

### 2.3. Detection of Carbapenemase Production

Among the five isolates that exhibited phenotypic resistance to carbapenems, two tested positive in the Blue-Carba assay, confirming the production of active carbapenemases. Molecular analysis identified both isolates—Pseudomonas fluorescens and Pseudomonas synxantha—as carriers of the bla_VIM_ gene, which encodes Verona integron-encoded metallo-β-lactamase (VIM). This enzyme is one of the most prevalent carbapenemases in the *Pseudomonas* genus and represents a serious global public health threat.

## 3. Discussion

AMR represents a complex global challenge that transcends clinical and veterinary contexts, increasingly impacting environmental health [21,22]. In this regard, aquatic ecosystems have been widely recognized as critical environments for the study of AMR, as they act simultaneously as reservoirs and vectors for the dissemination of ARB and ARGs [23,24,25,26,27]. This study characterized ARB isolated from the Aconcagua River, a freshwater system located in central Chile, which, due to intensive agricultural, livestock, aquaculture, and mining activities, as well as the discharge of urban wastewater, constitutes a representative model for environmental AMR monitoring. One of the most noteworthy findings was the effectiveness of the MALDI-TOF system in the taxonomic identification of environmental bacteria, enabling the species-level assignment of 76.9% of isolates (80/104), including many genera not traditionally associated with human health. This capability highlights the growing utility of this technique for environmental microbiology studies, even though commercial databases remain predominantly enriched with clinically relevant strains [28]. The accurate identification of environmental bacteria by MALDI-TOF expands the potential for understanding microbial ecology in human-impacted settings [28].

Another relevant observation was the absence of World Health Organization (WHO)-listed priority pathogens. While this could be interpreted as a favorable indicator from a public health perspective, it may also reflect methodological limitations in recovering certain genera, particularly fastidious or clinically relevant strains that require specific culture conditions. However, it is important to recognize that this result is not solely attributable to methodological constraints—it also reflects the current state of the microbial community in the river and the environmental conditions prevailing at the time of sampling. To address these temporal limitations and better understand seasonal dynamics, a longitudinal study is necessary to assess the temporal variation in ARB and ARGs across different seasons in the Aconcagua River.

On the other hand, there was a marked predominance of the genus *Pseudomonas*, which accounted for 40 of the 85 identified isolates—an observation that aligns with previous reports describing *Pseudomonas* as a highly adaptable and prevalent genus in contaminated aquatic environments, capable of tolerating extreme conditions and diverse toxic compounds, including heavy metals and organic pollutants [29,30]. The exclusive recovery of *Pseudomonas* and *Aeromonas* isolates on media supplemented with ceftazidime (CAZ), along with the absence of growth on ciprofloxacin (CIP)-supplemented plates, suggests a differential selective pressure potentially linked to the contaminant profile of the basin. This pattern may reflect the persistence of resistance determinants to third-generation cephalosporins relative to fluoroquinolones, whose resistance mechanisms often impose a high physiological cost [31,32,33]. In fact, 92.5% of isolates recovered on CAZ media exhibited concurrent resistance to aztreonam (ATM), raising additional concerns regarding the role of these genera as reservoirs of clinically relevant ARGs.

The detection of carbapenem resistance in five *Pseudomonas* isolates—all classified as environmental species—underscores the expansion of critical resistance mechanisms beyond hospital settings. Carbapenems are considered last-resort antibiotics for treating severe infections [21,34], and their resistance in non-pathogenic strains signals the possible environmental dissemination of high-priority resistance genes such as *bla_VIM_*. This hypothesis is further supported by the phenotypic confirmation of carbapenemase activity and molecular detection of *bla_VIM_* in two *Pseudomonas* isolates, highlighting the role of the river as an active reservoir of resistance determinants with high clinical relevance. While VIM is the most frequently reported carbapenemase in *Pseudomonas aeruginosa* [35,36,37,38], its presence in non-pathogenic members of the genus may facilitate its spread in natural environments. A likely explanation for the detection of *bla_VIM_* in this study is the high prevalence of *Pseudomonas* spp. in the sampled sites, a genus known for its intrinsic and acquired resistance mechanisms and for acting as a reservoir of mobile genetic elements. Furthermore, the two VIM-positive isolates were recovered from sites P5 and P8, both located immediately downstream of urban wastewater treatment plants (WWTPs). These sites are subject to high anthropogenic pressure and urban influence, which increases the likelihood of the environmental introduction of resistance genes originating from clinical sources. Therefore, it is plausible that *bla_VIM_* was introduced into the river system through the discharge of treated or partially treated hospital or domestic effluents, followed by its persistence and dissemination among environmental bacteria.

Geographically, the sampling sites P5 and P8—located immediately downstream of the WWTPs of Los Andes and Quillota, respectively—showed the highest concentrations of ARB, reinforcing the association between treated effluent discharge and the environmental load of resistant bacteria [27,39,40,41]. This finding underscores the urgent need to revise Chile’s wastewater treatment standards, which currently lack microbiological criteria, thereby permitting the potential release of ARB and ARGs into natural water bodies.

From a One Health perspective, the findings of this study are of particular concern. The waters of the Aconcagua River are extensively used for agricultural irrigation in one of Chile’s most important horticultural regions. The presence of ARB and ARGs in these waters poses a tangible risk for the introduction of resistant microorganisms into the food chain, with potential impacts on both human and animal health [42].

In conclusion, although WHO priority pathogens were not detected, the high prevalence of resistant *Pseudomonas*, the presence of carbapenem-resistant strains, and the detection of clinically significant carbapenemases such as VIM point to a worrisome environmental health scenario. These findings emphasize the need to establish integrated microbiological surveillance systems and implement public policies aimed at mitigating the environmental dissemination of AMR, particularly in river basins under strong anthropogenic pressure such as the Aconcagua River.

## 4. Materials and Methods

### 4.1. Study Area and Sampling

The study area corresponds to the Aconcagua River, located in the Valparaíso Region of Chile. The river stretches 177 km, originating in the Andes Mountains (−32.827014, −70.092600) and flowing westward until it empties into the Pacific Ocean at the city of Concón (−32.914798, −71.508738). The river basin covers an area of approximately 7340 km^2^. The estimated annual net discharge is 429 million m^3^, with an average flow rate of around 13.6 m^3^/s. Thirteen sampling points were selected along the course of the river (Figure 4), and all samples were collected on the same day. At each site, one liter of surface water was collected by submerging a sterile bottle 30 cm below the surface and opening it underwater. The samples were transported under cold chain conditions on the same day to the Molecular Genetics Laboratory at the Pontificia Universidad Católica de Valparaíso.

### 4.2. Processing and Isolation of Antibiotic-Resistant Bacteria (ARB)

Each water sample was passed through a 0.22 μm filter. The filters were resuspended in 0.85% saline solution, and serial 1:10 dilutions were prepared from 10^−1^ to 10^−4^. From each dilution, 100 μL was plated onto MacConkey agar plates independently supplemented with either ceftazidime (CAZ) or ciprofloxacin (CIP), both at a final concentration of 2 μg/mL. Cycloheximide was added at a concentration of 50 μg/mL to inhibit fungal growth. A control plate without antibiotics was also included. All plates were incubated at 37 °C for 24 h. Colonies with distinct morphologies—based on size, color, and shape—were selected from plates showing isolated growth. To ensure the purity of each morphotype, the selected colonies were re-streaked onto tryptic soy agar (TSA) plates. The purified colonies were then preserved in 12% glycerol at −80 °C. Morphotype identification was performed using MALDI-TOF mass spectrometry with the Bruker Biotyper^®^ system (Bruker, Billerica, MA, USA). Isolates with a score value ≥ 2.000 were identified at the species level, while those with scores between 1.700 and 1.999 could only be identified at the genus level. Isolates with score values below 1.700 could not be assigned to any genus [43,44].

### 4.3. Antibiotic Susceptibility Testing

The antibiotic susceptibility of the isolated bacteria was evaluated using the Kirby–Bauer disk diffusion method, in accordance with Clinical and Laboratory Standards Institute (CLSI) guidelines [45]. Only isolates that grew on plates supplemented with the antibiotics CAZ and CIP were subjected to susceptibility testing. Antibiotic panels were selected based on genus-specific recommendations: *Pseudomonas* spp. were tested using agents specified in CLSI M100, 2023 [45], and *Aeromonas* spp. were tested according to CLSI M45, 2016 [46]. The antibiotics tested included ceftazidime (CAZ, 30 μg), cefepime (FEP, 30 μg), ceftriaxone (CRO, 30 μg), aztreonam (ATM, 30 μg), imipenem (IPM, 10 μg), meropenem (MEM, 10 μg), piperacillin/tazobactam (TZP, 100/10 μg), ciprofloxacin (CIP, 5 μg), levofloxacin (LEV, 5 μg), gentamicin (CN, 10 μg), amikacin (AMK, 30 μg), trimethoprim/sulfamethoxazole (SXT, 1.25/23.75 μg), and tetracycline (TE, 30 μg). Plates were incubated at 37 °C for 24 h, and inhibition zones were interpreted according to CLSI breakpoints. Isolates were classified as susceptible (S), intermediate (I), or resistant (R). Where applicable, isolates were further categorized as multidrug-resistant (MDR), extensively drug-resistant (XDR), or pandrug-resistant (PDR) based on standardized international criteria [47]. Escherichia coli ATCC 25,922 was used as a quality control strain for all antibiotic susceptibility testing procedures.

### 4.4. Detection of Carbapenemase Production Using the Blue-Carba Test

Each isolate exhibiting resistance to at least one carbapenem was streaked onto an agar plate and incubated overnight at 37 °C. The following day, a single colony was selected and resuspended in 200 µL of sterile saline solution, adjusting the turbidity to a 5 McFarland standard. Subsequently, 100 µL of this bacterial suspension was mixed with 100 µL of the Blue-Carba master mix, composed of 0.04% (*w*/*v*) bromothymol blue (pH 6.8), 1 mM zinc sulfate (ZnSO_4_·7H_2_O), and 600 µg/mL imipenem. The mixture was dispensed into individual wells in a sterile 96-well microtiter plate. The plate was incubated at 37 °C, and the results were visually assessed after 90 min. A positive result, indicative of carbapenemase activity, was defined by a color shift from blue to green or yellow, resulting from acidification due to imipenem hydrolysis. Wells that retained the original blue coloration were considered negative. A KPC-producing Klebsiella pneumoniae strain was included as a positive control, while *E. coli* ATCC 25,922 was used as the negative control [48].

### 4.5. Genotypic Detection of Clinically Relevant Carbapenemases

Isolates that tested positive in the Blue-Carba assay were further screened by PCR for clinically important carbapenemase genes (KPC, VIM, NDM, and IMP). For this purpose, isolates were grown overnight in tryptic soy broth (TSB) supplemented with 4 µg/mL of imipenem. One milliliter of the culture was centrifuged at 10,000× *g* to obtain a cell pellet. Genomic DNA was extracted using Chelex resin. PCR amplification was performed using GoTaq^®^ Green Master Mix (Promega, Madison, WI, USA). Reactions were carried out in a final volume of 25 μL, containing 12.5 μL of master mix, 1 μL of each primer (10 μM), 2 μL of DNA template, and nuclease-free water to a final volume of 25 μL. The thermal cycling conditions consisted of an initial denaturation at 95 °C for 10 min, followed by 30 cycles of denaturation at 95 °C for 30 s, annealing at 55 °C for bla_KPC_ and bla_IMP_ or 60 °C for bla_VIM_ and bla_NDM_ for 30 s, and extension at 72 °C for 1 min (bla_KPC_, 798 bp), 30 s (bla_VIM_, 504 bp; bla_NDM_, 452 bp), or 15 s (bla_IMP_, 188 bp), with a final extension step at 72 °C for 10 min [49,50,51]. Reactions were carried out using a C1000 Touch Thermal Cycler (BioRad, Hercules, CA, USA). As positive controls, TOPO-TA plasmids carrying bla_KPC_, bla_VIM_, bla_NDM_, and bla_IMP_ independently were used. All conditions are detailed in Appendix A. Amplification products were visualized by electrophoresis on 1% agarose gels.

## Figures and Tables

**Figure 1 antibiotics-14-00669-f001:**
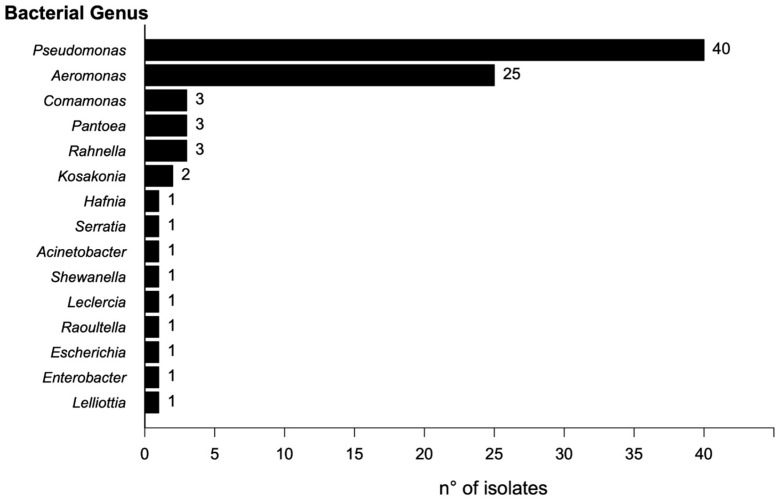
Distribution of bacterial genera isolated from the river identified by MALDI-TOF.

**Figure 2 antibiotics-14-00669-f002:**
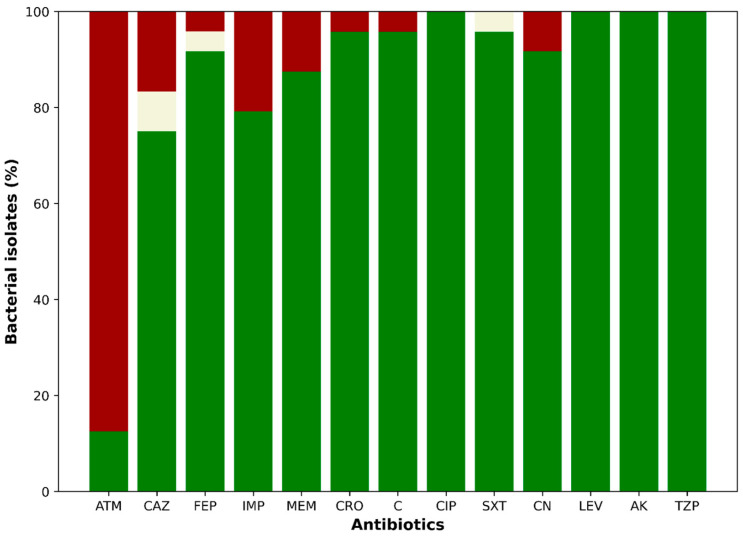
**Antibiotic susceptibility profiles of bacterial isolates recovered from ceftazidime-supplemented media**. Bars represent the percentage of isolates classified as susceptible (green), resistant (red), or intermediate (yellow) according to CLSI guidelines. Abbreviations: ATM, aztreonam; CAZ, ceftazidime; FEP, cefepime; IMP, imipenem; MEM, meropenem; CRO, ceftriaxone; C, chloramphenicol; CIP, ciprofloxacin; SXT, trimethoprim/sulfamethoxazole; CN, gentamicin; LEV, levofloxacin; AK, amikacin; TZP, piperacillin/tazobactam. Green: Susceptible; red: resistant; yellow: intermediate.

**Figure 3 antibiotics-14-00669-f003:**
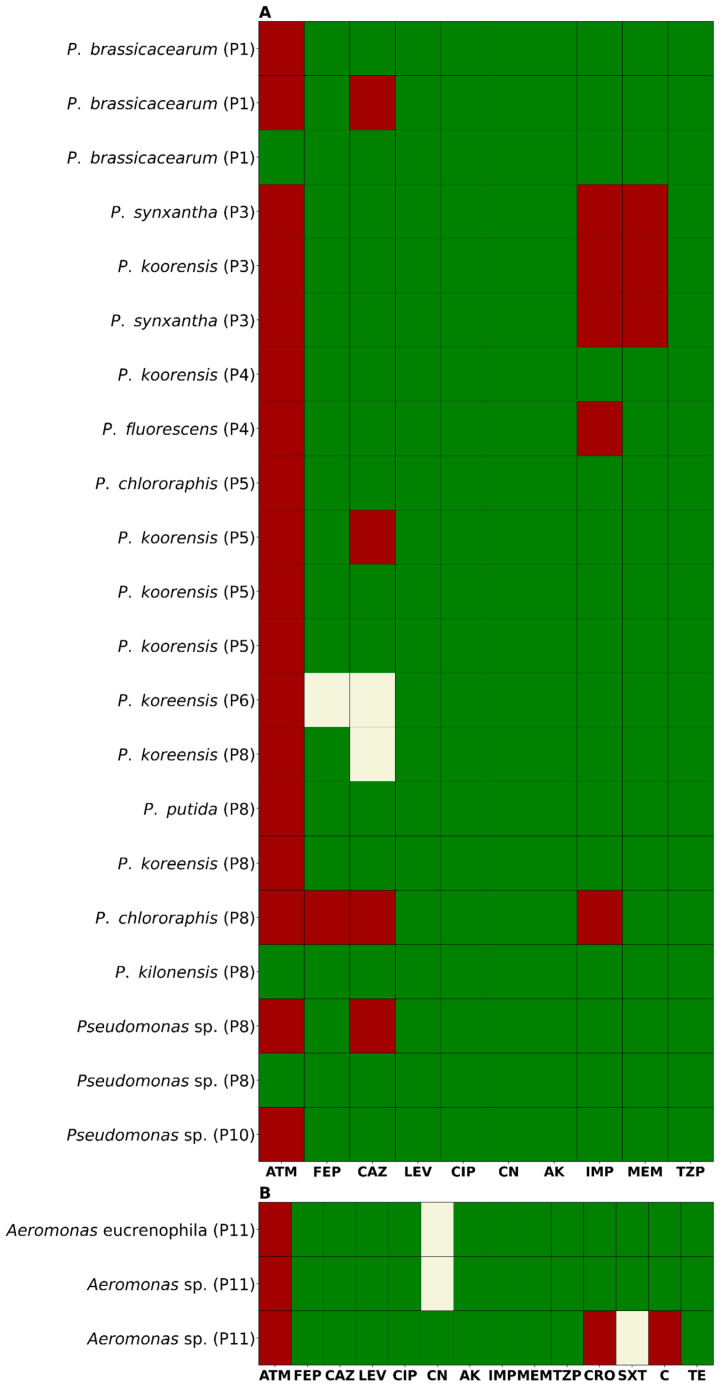
**Antibiotic susceptibility profiles of selected bacterial isolates.** (**A**) Pseudomonas spp. isolates; (**B**) Aeromonas spp. isolates. Bacterial isolates were identified by MALDI-TOF and its sampling point (P1–P11). Colors indicate susceptibility classification based on CLSI guidelines: green (susceptible), yellow (intermediate), and red (resistant). Antibiotic abbreviations: ATM, aztreonam; FEP, cefepime; CAZ, ceftazidime; LEV, levofloxacin; CIP, ciprofloxacin; CN, gentamicin; AK, amikacin; IMP, imipenem; MEM, meropenem; TZP, piperacillin/tazobactam; CRO, ceftriaxone; SXT, trimethoprim/sulfamethoxazole; C, chloramphenicol; TE, tetracycline. Green: Susceptible; red: resistant; yellow: intermediate.

**Figure 4 antibiotics-14-00669-f004:**
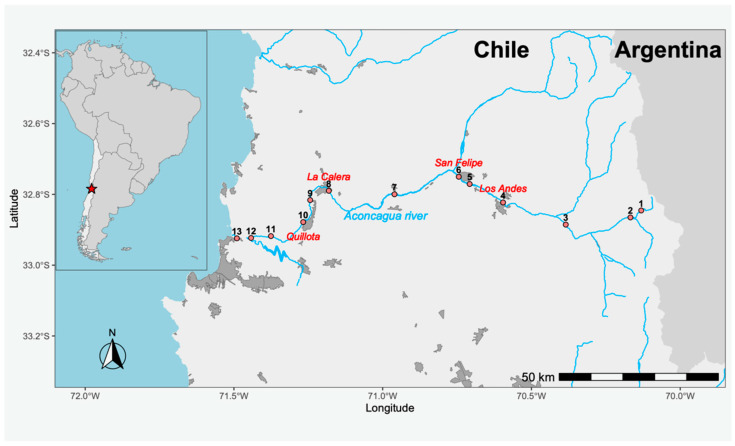
**Map of the Aconcagua River in central Chile.** Red dots indicate the sampling sites. The labeled cities are those with over 50,000 inhabitants through which the river directly flows.

## Data Availability

Data are contained within the article and Appendix A.

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
