# Peer review of "Antimicrobial Resistance in the Aconcagua River, Chile: Prevalence and Characterization of Resistant Bacteria in a Watershed Under High Anthropogenic Contamination Pressure"

_antibiotics, 2025, doi:10.3390/antibiotics14070669_

Round 1
Reviewer 1 Report
Comments and Suggestions for Authors
The study aims to isolate and characterize antibiotic-resistant bacteria along the Aconcagua River in Chile. A total of 104 bacterial isolates were identified to the genus and species level using MALDI-TOF mass spectrometry. Among these, Pseudomonas (n = 40) and Aeromonas (n = 25) were the most prevalent. Three isolates each belonged to the genera Comamonas, Rahnella, Pantoea, and Kosakonia. The antimicrobial resistance profile revealed that 87.5% of isolates were resistant to aztreonam. Additionally, five isolates were resistant to carbapenems, and two of them tested positive for carbapenemase activity and harbored the blaVIM gene.
This work is highly relevant in the context of One Health and contributes to our understanding of antimicrobial resistance (AMR) in aquatic environments. The sampling design and focus on environmental surveillance are commendable. However, the manuscript requires major revisions to improve its suitability for publication.
Comment for authors
- Please specify the statistical methods used to calculate proportions, generate graphs, and interpret the sampling data used?
- The descriptions of the Blue-Carba test and PCR protocols are currently too brief. Please elaborate on the methodology,
- The AST test was done without quality control? Please add them if you used it
- Ensure that all organism names are correctly italicized and used consistently throughout the manuscript.
- The emergence of the blaVIM gene in isolates from the Aconcagua River is a significant finding. Please expand the discussion to explore potential sources and transmission routes
- Line 293: Please listing all abbreviations used in the manuscript
- Please remove table 1 and rewrite the information properly
Author Response
Response to Reviewer 1
We would like to begin by sincerely thanking Reviewer 1 for their valuable comments and suggestions. Below, we provide point-by-point responses to each of the observations raised:
General comment:
This study aimed to isolate and characterize antibiotic-resistant bacteria along the Aconcagua River in Chile. A total of 104 bacterial isolates were identified to the genus and species levels using MALDI-TOF mass spectrometry. Among these, Pseudomonas (n = 40) and Aeromonas (n = 25) were the most prevalent. Three isolates each belonged to the genera Comamonas, Rahnella, Pantoea, and Kosakonia. The antimicrobial resistance profile revealed that 87.5% of the isolates were resistant to aztreonam. Additionally, five isolates were resistant to carbapenems, two of which tested positive for carbapenemase activity and harbored the blaVIM gene. This work is highly relevant within the One Health framework and makes a significant contribution to the understanding of antimicrobial resistance (AMR) in aquatic environments. The sampling design and the focus on environmental surveillance are particularly commendable. However, the manuscript requires substantial revisions to improve its suitability for publication.
Comment 1: Please specify the statistical methods used to calculate proportions, generate graphs, and interpret the sampling data used.
Response: We appreciate this comment. In this study, results are presented as simple proportions, calculated as the number of resistant isolates divided by the total number of identified isolates. No additional statistical analyses were conducted, as they were not considered necessary for this descriptive analysis.
Comment 2: The descriptions of the Blue-Carba test and PCR protocols are currently too brief. Please elaborate on the methodology.
Response: We have expanded the Materials and Methods section to include detailed descriptions of the Blue-Carba test procedure and the PCR conditions.
Comment 3: The AST test was done without quality control? Please add them if you used it.
Response: Information regarding the use of a positive control strain in antimicrobial susceptibility testing (AST) has been added to the revised manuscript.
Comment 4: Ensure that all organism names are correctly italicized and used consistently throughout the manuscript.
Response: We thoroughly revised the manuscript to ensure that all genus and species names are consistently and correctly italicized.
Comment 5: The emergence of the blaVIM gene in isolates from the Aconcagua River is a significant finding. Please expand the discussion to explore potential sources and transmission routes.
Response: We appreciate this insightful observation and have expanded the discussion to address potential sources and transmission routes of the blaVIM gene. Based on current data, we hypothesize that the inadequate operation of wastewater treatment plants may play a role. We refrain from further speculation beyond the available evidence. Robust spatial epidemiological studies are needed to better understand the transmission routes involved.
Comment 6: Line 293: Please list all abbreviations used in the manuscript.
Response: A comprehensive list of all abbreviations used in the manuscript has been included, as requested.
Comment 7: Please remove Table 1 and rewrite the information properly.
Response: The information from Table 1 has been rewritten and integrated into the main text. Additionally, the original table has been relocated to the supplementary material section.

Reviewer 2 Report
Comments and Suggestions for Authors
Major comments
The authors should provide more information on the reaction mixtures and conditions for the PCR in the materials and methods for genotypic resistance identification. They must also indicate the instruments used.
Ensure scientific names are properly written, with the genus and species names in italics.
Figure 2 repeats Lines 108-125. Choose one way to represent this data: either as a paragraph or as a graph. If the graph is chosen, it must only point the reader to the key findings and let the Figure show the rest.
The authors mention several times that WHO priority pathogens were not isolated in their study, and attribute this only to methodological issues. It would be good for them to indicate that the once-off sampling only indicated the potential role of the river as a reservoir for ARB. This might have been different if they had sampled several times and included seasonal variations. With a once-off sampling, a hit does not automatically mean a problem, just as a miss cannot mean the absence of the problem. So, the authors may want to indicate their sampling as a limitation and also indicate its potential effect on the absence of the WHO priority organism. This should also be included in the conclusion.
Specific comments
Line 45: Delete "The" at the start of the sentence.
Line 46: Delete "being"
Line 55: Delete "true". I do not think there are "false" emerging contaminants
Line 57: Which elements do the authors refer to here? Are ARB also horizontally transferred?
Author Response
Response to Reviewer 2:
We sincerely appreciate the valuable comments provided by Reviewer 2. Below, we address each of the points raised:
Major Comments
Comment 1: The authors should provide more information on the reaction mixtures and conditions for the PCR in the materials and methods for genotypic resistance identification. They must also indicate the instruments used.
Response: We thank the reviewer for this observation. In the revised manuscript, we have included the requested information detailing the reaction mixtures, PCR conditions, and the equipment used, including the brand and model of the thermal cycler.
Comment 2: Ensure scientific names are properly written, with the genus and species names in italics.
Response: The manuscript was carefully reviewed, and all scientific names have been corrected to ensure proper italicization of genus and species.
Comment 3: Figure 2 repeats Lines 108–125. Choose one way to represent this data: either as a paragraph or as a graph. If the graph is chosen, it must only point the reader to the key findings and let the Figure show the rest.
Response: We appreciate and value this comment. However, we believe that the information presented in the text and in Figure 2 is complementary rather than redundant. Additionally, we consider that the figure legend should remain as is, as figures should be self-explanatory and capable of conveying key information independently of the main text.
Comment 4: The authors mention several times that WHO priority pathogens were not isolated in their study, and attribute this only to methodological issues. It would be good for them to indicate that the once-off sampling only indicated the potential role of the river as a reservoir for ARB. This might have been different if they had sampled several times and included seasonal variations. With a once-off sampling, a hit does not automatically mean a problem, just as a miss cannot mean the absence of the problem. So, the authors may want to indicate their sampling as a limitation and also indicate its potential effect on the absence of the WHO priority organism. This should also be included in the conclusion.
Response: We are grateful for this suggestion and have fully incorporated it into the discussion and conclusion sections of the manuscript. We agree that a single sampling event represents an important methodological limitation. Nevertheless, we believe the results provide a representative snapshot of the river’s microbiota at the time of sampling. We have also included in the text the need for longitudinal studies to better understand how seasonal variations and other factors influence the dynamics of antimicrobial resistance in the Aconcagua River.
Specific Comments
Comment 1 (Line 45): Delete "The" at the start of the sentence.
Response: Corrected.
Comment 2 (Line 46): Delete "being".
Response: Corrected.
Comment 3 (Line 55): Delete "true". I do not think there are "false" emerging contaminants.
Response: Corrected.
Comment 4 (Line 57): Which elements do the authors refer to here? Are ARB also horizontally transferred?
Response: This sentence has been revised to clarify that the elements transferred horizontally are antimicrobial resistance genes (ARGs), not the antimicrobial-resistant bacteria (ARB).

Round 2
Reviewer 1 Report
Comments and Suggestions for Authors
I have carefully reviewed the revised manuscript. The authors have addressed my concerns thoroughly, and I have no further comments. The manuscript is acceptable in its current form.
Reviewer 2 Report
Comments and Suggestions for Authors
Thank you for addressing the comments